# Single-dose BNT162b2 vaccine protects against asymptomatic SARS-CoV-2 infection

Nick K Jones[1,2†], Lucy Rivett[1,2†], Shaun Seaman[3], Richard J Samworth[4], Ben Warne[1,5,6], Chris Workman[7], Mark Ferris[7], Jo Wright[7], Natalie Quinnell[7], Ashley Shaw[1], Cambridge COVID-19 Collaboration[1,6], Ian G Goodfellow[8], Paul J Lehner[1,5,6], Rob Howes[9], Giles Wright[7], Nicholas J Matheson[1,5,6,10‡], Michael P Weekes[1,6,11‡*]

[1]Cambridge University NHS Hospitals Foundation Trust, Cambridge, United Kingdom; [2]Clinical Microbiology & Public Health Laboratory, Public Health England, Cambridge, United Kingdom; [3]Medical Research Council Biostatistics Unit, University of Cambridge, Cambridge, United Kingdom; [4]Statistical Laboratory, Centre for Mathematical Sciences, University of Cambridge, Cambridge, United Kingdom; [5]Cambridge Institute of Therapeutic Immunology & Infectious Disease, University of Cambridge, Cambridge, United Kingdom; [6]Department of Medicine, University of Cambridge, Cambridge, United Kingdom; [7]Occupational Health and Wellbeing, Cambridge Biomedical Campus, Cambridge, United Kingdom; [8]Division of Virology, Department of Pathology, University of Cambridge, Cambridge, United Kingdom; [9]Cambridge COVID-19 Testing Centre and AstraZeneca, Anne McLaren Building, Cambridge, United Kingdom; [10]NHS Blood and Transplant, Cambridge, United Kingdom; [11]Cambridge Institute for Medical Research, University of Cambridge, Cambridge, United Kingdom

*For correspondence:
mpw1001@cam.ac.uk

†These authors contributed equally to this work
‡These authors also contributed equally to this work

Group author details:
Cambridge COVID-19 Collaboration See page 5

**Abstract** The BNT162b2 mRNA COVID-19 vaccine (Pfizer-BioNTech) is being utilised internationally for mass COVID-19 vaccination. Evidence of single-dose protection against symptomatic disease has encouraged some countries to opt for delayed booster doses of BNT162b2, but the effect of this strategy on rates of asymptomatic SARS-CoV-2 infection remains unknown. We previously demonstrated frequent pauci- and asymptomatic SARS-CoV-2 infection amongst healthcare workers (HCWs) during the UK's first wave of the COVID-19 pandemic, using a comprehensive PCR-based HCW screening programme (Rivett et al., 2020; Jones et al., 2020). Here, we evaluate the effect of first-dose BNT162b2 vaccination on test positivity rates and find a fourfold reduction in asymptomatic infection amongst HCWs ≥12 days post-vaccination. These data provide real-world evidence of short-term protection against asymptomatic SARS-CoV-2 infection following a single dose of BNT162b2 vaccine, suggesting that mass first-dose vaccination will reduce SARS-CoV-2 *transmission*, as well as the burden of COVID-19 *disease*.

## Introduction

The UK has initiated mass COVID-19 immunisation, with healthcare workers (HCWs) given early priority because of the potential for workplace exposure and risk of onward transmission to patients. The UK's Joint Committee on Vaccination and Immunisation has recommended maximising the number of people vaccinated with first doses at the expense of early booster vaccinations, based on

single-dose efficacy against symptomatic COVID-19 disease (*Department of Health and Social Care, 2021*; *Polack et al., 2020*; *Voysey et al., 2021*).

At the time of writing, three COVID-19 vaccines have been granted emergency use authorisation in the UK, including the BNT162b2 mRNA COVID-19 vaccine (Pfizer-BioNTech). A vital outstanding question is whether this vaccine prevents asymptomatic as well as symptomatic SARS-CoV-2 infection or merely converts infections from symptomatic to asymptomatic. Sub-clinical infection following vaccination could continue to drive transmission. This is especially important because many UK HCWs have received this vaccine, and nosocomial COVID-19 infection has been a persistent problem.

Through the implementation of a 24 hour turnaround PCR-based comprehensive HCW screening programme at Cambridge University Hospitals NHS Foundation Trust (CUHNFT), we previously demonstrated the frequent presence of pauci- and asymptomatic infection amongst HCWs during the UK's first wave of the COVID-19 pandemic (*Rivett et al., 2020*). Here, we evaluate the effect of first-dose BNT162b2 vaccination on test positivity rates and cycle threshold (Ct) values in the asymptomatic arm of our programme, which now offers weekly screening to all staff.

## Results and discussion

Vaccination of HCWs at CUHNFT began on 8 December 2020, with mass vaccination from 8 January 2021. Here, we analyse data from 2 weeks spanning 18–31 January 2021, during which (1) the prevalence of COVID-19 amongst HCWs remained approximately constant and (2) we screened comparable numbers of vaccinated and unvaccinated HCWs. During this period, 4408 (week 1) and 4411 (week 2) PCR tests were performed on individuals reporting well to work, from a weekly on-site HCW population of ~9000. We stratified HCWs <12 days or ≥12 days post-vaccination because this was the point at which protection against symptomatic infection began to appear in the phase III clinical trial (*Polack et al., 2020*). In the post-vaccination groups, the median number of days between vaccination and testing were 7 (interquartile range [IQR] 4–9; <12 day group) and 16 (14–18; ≥12 day group).

Twenty-six of 3252 (0.8%, Wilson's interval 0.6–1.2%) tests from unvaccinated HCWs were positive (Ct < 36), compared to 13/3535 (0.4%, Wilson's interval 0.2–0.6%) tests from HCWs <12 days post-vaccination and 4/1989 (0.2%, Wilson's interval 0.1–0.5%) tests from HCWs ≥12 days post-vaccination (p=0.023 and p=0.004, respectively; Fisher's exact test, *Figure 1* and *Table 1*). This suggests a fourfold decrease in the risk of asymptomatic SARS-CoV-2 infection amongst HCWs ≥12 days post-vaccination, compared to unvaccinated HCWs, with an intermediate effect amongst HCWs <12 days post-vaccination.

A marked reduction in infections was also seen when analyses were repeated with (1) inclusion of HCWs testing positive through both the symptomatic and asymptomatic arms of the programme (56/3370 [1.7%, Wilson's interval 1.3–2.2%] unvaccinated vs 8/2018 [0.4%, Wilson's interval 0.2–0.8%] ≥12 days post-vaccination, 4.2-fold reduction, p<0.0001) and (2) inclusion of PCR tests that were positive at the limit of detection (Ct > 36, 42/3268 [1.3%, Wilson's interval 1.0–1.7%] vs 15/2000 [0.7%, Wilson's interval 0.5–1.2%], 1.7-fold reduction, p=0.07). In addition, the median Ct value of positive tests showed a non-significant trend towards increase between unvaccinated HCWs and HCWs ≥12 days post-vaccination (23.3 [IQR 13.5–33.0] to 30.3 [IQR 25.5–35.1], *Figure 1*), raising the possibility that vaccinated individuals who do go on to develop infection may have *lower* viral loads.

HCWs working in COVID-19 clinical areas were prioritised for vaccination, and a small number of clinically vulnerable HCWs were also given priority. Otherwise, vaccine allocation was arbitrary. Since asymptomatic infection was examined, the date of infection could have been earlier than the test date. These factors would all tend to lead to an underestimate of the vaccine's effect (bias towards the null). Because of the rapid decline in the incidence of SARS-CoV-2 infection in the Cambridge community, this study could only examine the short-term impact of single-dose BNT162b2 vaccination. The frequency of prior SARS-CoV-2 infection (*Cooper et al., 2020*) was similar in all groups (seroprevalence 7.1%, unvaccinated; 5.6%, <12 days post-vaccination; 5.7%, ≥12 days post-vaccination), suggesting that this did not confound our observations.

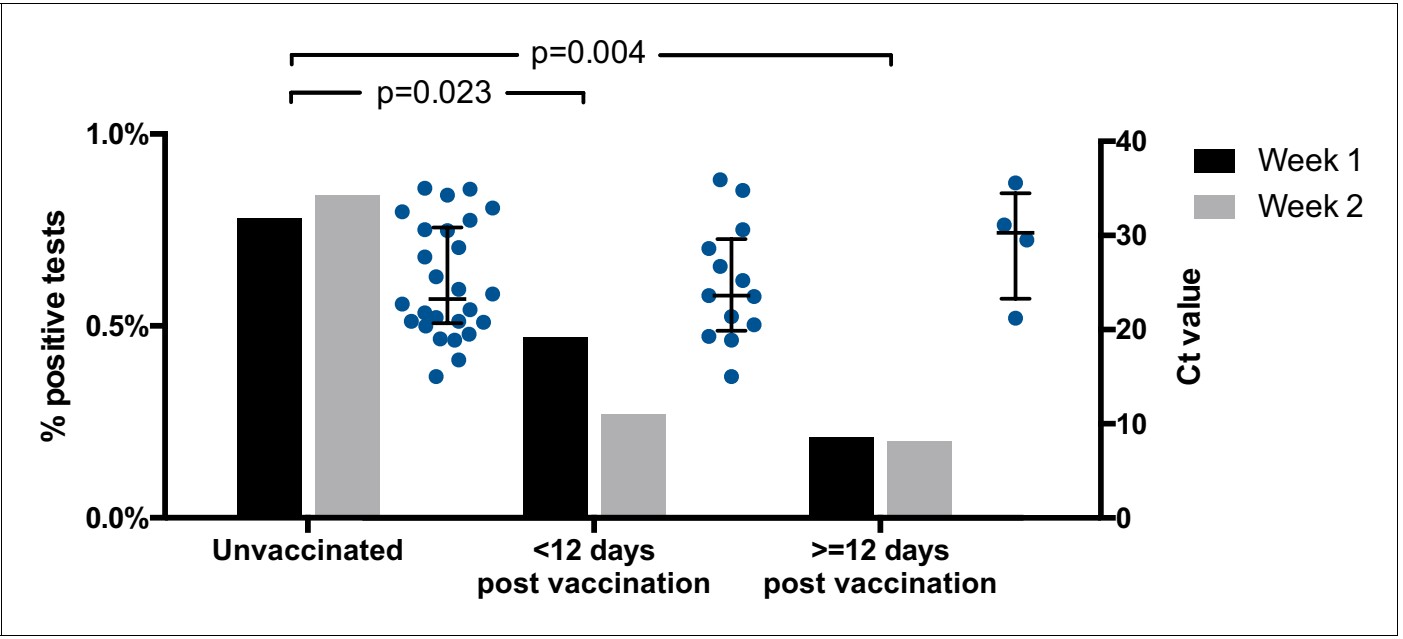

**Figure 1.** Proportion of positive screening tests for SARS-CoV-2 amongst HCWs from the CUHNHFT asymptomatic screening programme (grey bars; week 1, 18–24 January 2021; week 2, 25–31 January 2021) and Ct values of positive tests (Ct < 36; blue dots; both weeks). RT-PCR targeting the SARS-CoV-2 ORF1ab genes was conducted at the Cambridge COVID-19 Testing Centre (part of the UK Lighthouse Labs Network). For proportions of positive screening tests, p-values for pair-wise comparisons of unvaccinated HCWs with HCWs <12 days or ≥12 days post-vaccination are shown (Fisher's exact test; both weeks). For Ct values, medians ± interquartile ranges are shown.

The online version of this article includes the following source data for figure 1:

**Source data 1.** Proportions of positive asymptomatic SARS-CoV-2 screens and distributions of Ct values.

Taken together, our findings provide real-world evidence of short-term protection against asymptomatic SARS-CoV-2 infection after a single dose of BNT162b2 vaccine, at a time when the UK COVID-19 variant of concern 202012/01 (lineage B.1.1.7) accounted for the great majority of infections (24/29 sequenced isolates from asymptomatic HCWs). A fourfold reduction from 0.8% to 0.2% in asymptomatic infection is likely to be crucial in controlling nosocomial SARS-CoV-2 transmission. Nonetheless, protection is incomplete, suggesting that continuing asymptomatic HCW screening, social distancing, mask-wearing, and strict hand hygiene remain vital.

**Table 1.** Weekly numbers and proportions of positive SARS-CoV-2 test results spanning 6 weeks around the main study period (indicated in grey).

| Week start | Unvaccinated | | | <12 Days since vaccination | | | ≥12 Days since vaccination | | |
| --- | --- | --- | --- | --- | --- | --- | --- | --- | --- |
| | Total tests | Positive tests | % | Total tests | Positive tests | % | Total tests | Positive tests | % |
| 28 December 2020 | 2097 | 16 | 0.8% | 8 | 0 | 0.0% | 6 | 0 | 0.0% |
| 4 January 2021 | 4762 | 43 | 0.9% | 93 | 0 | 0.0% | 22 | 0 | 0.0% |
| 11 January 2021 | 3273 | 27 | 0.8% | 978 | 6 | 0.6% | 30 | 0 | 0.0% |
| 18 January 2021 | 2183 | 17 | 0.8% | 1716 | 8 | 0.5% | 483 | 1 | 0.2% |
| 25 January 2021 | 1069 | 9 | 0.8% | 1819 | 5 | 0.3% | 1506 | 3 | 0.2% |
| 1 February 2021 | 699 | 1 | 0.1% | 758 | 1 | 0.1% | 2825 | 1 | 0.0% |

## Materials and methods

### HCW screening programme

We previously described protocols for staff screening, sample collection, and results reporting in detail (*Rivett et al., 2020*; *Jones et al., 2020*). In general, these methods remained unchanged throughout this study period. Two parallel streams of entry into the testing programme included (1) *HCW symptomatic* and *HCW symptomatic household contact* screening arms and (2) an *HCW asymptomatic* screening arm. Since our prior description of the screening programme, weekly asymptomatic testing is now offered to all CUHNFT staff. Testing was performed (1) at temporary on-site 'Pods' and (2) via self-swabbing kits collected by HCWs. Individuals performed a self-swab of the oropharynx and anterior nasal cavity. Samples were subjected to RNA extraction and amplification using real-time RT-PCR, with all sample processing and analysis undertaken at the Cambridge COVID-19 Testing Centre (Lighthouse Laboratory).

### Vaccination

HCW vaccination began at CUHNFT on 8 December 2020, with appointments made by invitation only for all high-risk HCWs working on-site. This was followed by self-booked appointments for HCWs working in designated COVID-19 clinical areas, from 8 January 2021 onwards. From 18 January 2021, vaccination was offered to all HCWs, with appointments made via a booking website and latterly using the hospital's electronic patient record system 'MyChart'. All vials of Pfizer-BioNTech COVID-19 Vaccine (BNT162b2) were stored at $-74°C$, before being transferred to storage at $2–8°C$. From the moment the vials were removed from the freezer, they were given a 120 hr expiration date, of which 3 hr were dedicated to thawing the vaccines. All vaccine doses were administered intramuscularly by trained vaccinators, in accordance with the manufacturer's instructions. Vaccination was undertaken exclusively at an on-site vaccination centre, with mandatory mask-wearing and social distancing in place. HCWs remained at the on-site vaccination centre for a minimum observation period of 15 min after vaccination.

### Data extraction and analysis

Swab result, vaccination details, and serology data for HCWs were extracted directly from the hospital-laboratory interface software, Epic (Verona, WI). Data were collated using Microsoft Excel and the figure produced with GraphPad Prism (GraphPad Software, La Jolla, CA). Fisher's exact test was used for the comparison of positive rates between groups, defined in the main text. Additionally, 95% confidence intervals were calculated using Wilson's method.

## Acknowledgements

This work was funded in part by Wellcome Senior Clinical Research Fellowships (Grant numbers 108070/Z/15/Z to MPW, 207498/Z/17/Z to IGG), a Wellcome Principal Research Fellowship to PJL (210688/Z/18/Z), an MRC Clinician Scientist Fellowship (MR/P008801/1) and NHSBT workpackage (WPA15-02) to NJM, EPSRC grants to RJS (EP/P031447/1,EP/N031938/1), and an MRC grant to SS (MC_UU_00002/10). The sequencing costs were funded by the COVID-19 Genomics UK (COG-UK) Consortium, which is supported by funding from the Medical Research Council (MRC) part of UK Research and Innovation (UKRI), the National Institute of Health Research (NIHR), and Genome Research Limited, operating as the Wellcome Sanger Institute. Funding was also received from Addenbrooke's Charitable Trust and the NIHR Cambridge Biomedical Research Centre. For the purpose of open access, the author has applied a CC BY public copyright licence to any Author Accepted Manuscript version arising from this submission. We also acknowledge contributions from all staff at CUHNFT Occupational Health and Wellbeing, the COVID-19 vaccination programme and the Cambridge COVID-19 Testing Centre.

## Additional information

### Group author details

**Cambridge COVID-19 Collaboration**
Amy Amory; Stephen Baker; Emma Bateman; Aklima Begum; Moushima Begum; John Bradley; Michael Brennan; Helen Burn; Caroline Crofts; Afzal Chaudhry; Yasmin Chaudhry; Daniel J Cooper; Sharon Dawson; Gordon Dougan; Renny Feather; Louise Free; Katie Friel; Claire Gildea; Iliana Georgana; Lizz Grimwade; Ravi Gupta; Susan Hall; Sophie Hannan; James Hayes; Aleksandra Hosaja; Myra Hosmillo; Rhys Izuagbe; Aminu Jahun; Lidia James; Jill Jardin; Nathalie Kingston; Sara Lear; Paul A Lyons; Patrick H Maxwell; Sue Mott; Sarah Mugavin; Joyce Mwiya; Sharon Peacock; Ravi Prakash Nallattil; Kazeem Oloyede; Willem H Ouwehand; Elle Page; Marina Perez; Tim Raine; Matthew Routledge; Caroline Saunders; Kenneth GC Smith; Dominic Sparkes; Maria Stafford; Charlotte Summers; Despiona Tatsi; James ED Thaventhiran; Sharon Thomas Johnson; M Estée Török; Mark Toshner; Lesley Turner; Kate Wall; Karis Watson

### Competing interests

Rob Howes: Dr Howes was employed by AstraZeneca PLC during the period of study and preparation of this manuscript. The other authors declare that no competing interests exist.

### Funding

| Funder | Grant reference number | Author |
| --- | --- | --- |
| Wellcome Trust | 108070/Z/15/Z | Michael P Weekes |
| Wellcome Trust | 207498/Z/17/Z | Ian G Goodfellow |
| Wellcome Trust | 210688/Z/18/Z | Paul J Lehner |
| Medical Research Council | MR/P008801/1 | Nicholas J Matheson |
| NHS Blood and Transplant | WPA15-02 | Nicholas J Matheson |
| EPSRC | EP/P031447/1 | Richard J Samworth |
| Medical Research Council | MC_UU_00002/10 | Shaun Seaman |
| EPSRC | EP/N031938/1 | Richard J Samworth |

The funders had no role in study design, data collection and interpretation, or the decision to submit the work for publication.

### Author contributions

Nick K Jones, Lucy Rivett, Data curation, Formal analysis, Investigation, Methodology, Writing - original draft, Project administration; Shaun Seaman, Richard J Samworth, Investigation, Methodology, Writing - review and editing; Ben Warne, Ian G Goodfellow, Formal analysis, Investigation, Writing - review and editing; Chris Workman, Data curation, Formal analysis, Project administration, Writing - review and editing; Mark Ferris, Investigation, Methodology, Project administration, Writing - review and editing; Jo Wright, Natalie Quinnell, Data curation, Investigation, Project administration, Writing - review and editing; Ashley Shaw, Supervision, Project administration, Writing - review and editing; Cambridge COVID-19 Collaboration, Resources, Data curation; Paul J Lehner, Conceptualization, Methodology, Writing - review and editing; Rob Howes, Data curation, Investigation, Methodology, Project administration, Writing - review and editing; Giles Wright, Supervision, Investigation, Methodology, Project administration, Writing - review and editing; Nicholas J Matheson, Conceptualization, Formal analysis, Investigation, Methodology, Writing - review and editing; Michael P Weekes, Conceptualization, Data curation, Formal analysis, Investigation, Methodology, Writing - original draft, Project administration

## Author ORCIDs

Nick K Jones ⬡ https://orcid.org/0000-0003-4475-7761
Lucy Rivett ⬡ http://orcid.org/0000-0002-2781-9345
Ian G Goodfellow ⬡ http://orcid.org/0000-0002-9483-510X
Paul J Lehner ⬡ http://orcid.org/0000-0001-9383-1054
Nicholas J Matheson ⬡ https://orcid.org/0000-0002-3318-1851
Michael P Weekes ⬡ https://orcid.org/0000-0003-3196-5545

## Ethics

Human subjects: This study was conducted as a service evaluation of the CUHNFT staff testing and vaccination services (CUHNFT clinical project ID ID3682). As a study of healthcare-associated infections, this investigation is exempt from requiring ethical approval under Section 251 of the NHS Act 2006 (see also the NHS Health Research Authority algorithm, available at http://www.hra-decision-tools.org.uk/research/, which concludes that no formal ethical approval is required).

## Decision letter and Author response

Decision letter https://doi.org/10.7554/eLife.68808.sa1
Author response https://doi.org/10.7554/eLife.68808.sa2

# Additional files

## Supplementary files

• Transparent reporting form

## Data availability

All data generated or analysed during this study are included in the manuscript and supporting files. Source data file has been provided for Figure 1.

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
