## [Decision Letter]

**Acceptance summary:**

The investigators expanded their healthcare worker (HCW) asymptomatic screening programme by offering weekly testing to all staff at Cambridge University Hospitals. They also rapidly rolled out vaccination of the staff with the BNT162b2 mRNA vaccine (Pfizer-BioNTech). This provided the opportunity to investigate whether the vaccination protects against infection (i.e., transmission), or just abrogates COVID-19 disease. This is particularly important, because the vaccine has been widely administered to HCWs, and nosocomial infections have blighted hospitals throughout the pandemic. Published clinical trials data have so far only addressed whether this vaccine prevents symptomatic COVID-19. Here, the investigators show that asymptomatic SARS-COV-2 infection is also markedly reduced in a real-world setting, with four-fold fewer infections 12 days post-vaccination onward. The findings are of great interest, supporting the role of single-dose BNT162b2 vaccination to reduce SARS-CoV-2 transmission. The findings are also applicable to non-healthcare settings, may guide healthcare policies.

---

## [Author Response]

[Editors' note: we include below the reviews that the authors received from another journal, along with the authors’ responses.]

Reviewer #1:This short communication describes an opportunistic analysis of results of regular screening PCR tests for SARS CoV2 among healthcare workers some of whom had received the Pfizer vaccine and shows a reduction in positive results most marked from 12 days post dose. The allocation of vaccine was (presumably) non-random so there is some potential for bias which is not acknowledged (although this might have favoured frontline workers first who might be expected to be more exposed).

We thank the reviewer for highlighting this important potential source of confounding in our observations. We had not previously discussed confounding due to the stated word limits for ‘Correspondence’ articles. We have now included a description of the potential limitations of the study in a brief paragraph on limitations:

“HCWs working in COVID-19 clinical areas were prioritised for vaccination first, and a small number of clinically vulnerable HCWs were also given priority. Otherwise, vaccine allocation was uniformly at random. As asymptomatic infection was examined, the date of infection could have been earlier than the test date. These would all likely lead to an underestimate of the vaccine’s effect (bias towards the null). This study could only examine the short-term impact of single-dose BNT162b2 vaccination.”

It would also be helpful to know what the uptake rate of regular screening tests for HCWs at this hospital was at the time of the study and the range of time between vaccination and testing among those included in the analysis.

We thank the reviewer for this useful suggestion. Uptake at the time of study was ~50% (4,408 – 4,411 weekly tests from a population of ~9,000 HCW who worked at least one shift in each week [lines 45-46]). We have now included median and interquartile range values for the number of days since vaccination in each of the post-vaccine groups (lines 48-50).

Reviewer #2:This is an interesting and important analysis. However, I don't think the evidence presented is sufficient to claim that, "…we provide real world evidence for a high level of protection against symptomatic SARS-CoV-2 infection after a single dose of BNT162b2 vaccine.". This might be overstated. There is no data to support an argument that the findings last over time and without a second dose of the vaccine at 3/52.

We thank the reviewer for highlighting this important study limitation. We have now included a brief paragraph on study limitations, which emphasises that “this study could only examine the short-term impact of single-dose BNT162b2 vaccination”. We have also re-worded the first sentence of the concluding paragraph, so as not to overstate the study conclusions: “Our findings provide real-world observational evidence of high level short-term protection against asymptomatic SARS-CoV-2 infection after a single dose of BNT162b2 vaccine, at a time of predominant transmission when the UK COVID-19 variant of concern 202012/01 (lineage B.1.1.7) accounted for >90% of local infections, and amongst a population with a relatively low frequency of prior infection (7.2% antibody positive).”.

Also, there is no evidence presented that I can see to suggest that the single dose vaccine is effective against the B.1.1.7 strain. Perhaps the authors could comment on the % of SARS-CoV-2 infections that were B.1.1.7?

We thank the reviewer for this useful suggestion. We have now included a sentence on the rate of B.1.1.7 lineage detection in samples from local community at the time of study “at a time when the UK COVID-19 variant of concern 202012/01 (lineage B.1.1.7) accounted for >90% of local infections”. This is based on publicly available COG-UK sequencing data, which found that from 228 sequenced samples from Cambridgeshire from between 18/01/21 and 31/01/21, 207 (90.8%) were identified as lineage B.1.1.7. https://β.microreact.org/project/kqW9xZtPytsXUpbyhVFCZM-cog-uk-2021-02-25-uk-sars-cov-2/

Reviewer #3:Thank you for the opportunity to review this timely manuscript. As the COVID vaccines start to make their way out of the highest-risk populations and into more generally at-risk groups information regarding immediate efficacy and arresting of spread is essential for shaping vaccine rollout policies. I'd like to offer some comments on these preliminary findings:1a. The authors' presentation of the three different groups (unvaccinated, those with fewer than 12 days post-vaccine and those with more than 12 days post vaccine) is accompanied by p-values which are testing the rate of positive PCR tests over all by group and then under various subsets. I do not see the statistical validity of these tests. This is not a random sample or a randomized trial. Is there evidence to suggest that the HCW were randomly selected or received vaccines that were not purposive in some way? Applying p-values to this data suggests that there is some potential for replicability where I do not think such a condition actually exists based on this design. If there is some evidence to suggest that the vaccination of HCW was random then it should appear in this document. The absence of randomization of any kind opens the door to argument about potential (and unexplored) confounding.

Many thanks for these comments. We have discussed at length our approach and data with two leading statisticians in Cambridge, who as a result have now been included as full authors. Shaun Seaman works at the MRC Biostatistics Unit, and his particular interests are in methodologies for drawing causal inferences from observation data and for handling selection biases. Professor Richard Samworth is the University of Cambridge Professor of Statistical Science and the Director of the University Statistical Laboratory. Both agree with our approach, and have been instrumental in writing this rebuttal. In response to the points raised:

a. P-values and confounding. P-values are frequently used in observational studies such as ours. It is standard practice when analysing observational studies to present p-values for differences between non-randomised groups and then to discuss potential confounding and how such confounding might be expected to bias the results. A p-value may be less than 0.05 because `treatment' or `exposure' (in this case, vaccination) has a causal effect on the outcome (in this case, testing positive), because of random chance (which is unlikely when p<0.05), or because of confounding bias. We had not yet discussed confounding due to the stated word limits for ‘Correspondence’ articles, however have now included a paragraph explicitly discussing limitations of our study. HCWs working in COVID-19 clinical areas were prioritised for vaccination first, and a small number of clinically vulnerable HCWs working elsewhere in the hospital were also given priority. Apart from that, vaccine allocation was uniformly at random. This would be likely to lead to an underestimate of the vaccine’s effect on asymptomatic infection, and we now directly reference this potential bias in lines 66-71 of our paragraph on study limitations.

In addition, the dates of infection would have been earlier than the dates of test, meaning that some positive tests in individuals who had been vaccinated >12 days may have been due to infections that occurred <12 days after vaccination. This would cause a bias towards underestimating the benefit of vaccination.

b. Randomisation and potential for replicability. It is not being claimed that the data originate from a random sample or randomised controlled trial. However, for our study, the data could be considered to be random in at least three different ways. First, Cambridge University Hospitals (CUH) could be thought of as randomly chosen hospitals from a population of similar hospitals. Had we randomly chosen different hospitals, we would have observed different numbers of positive tests in the three vaccination groups. Second, we could think of the CUH staff as being a random sample from the population of possible staff that CUH might have employed. Again, had the staff been different, so would have been the numbers of positive tests in each of the groups. Third, we can imagine hypothetically re-running January 2021. Again, the numbers of positive tests in each of the groups would then be different. This conceptualisation of the sample as coming from a hypothetical infinite population is standard in statistics. See, for example, Professor David Spiegelhalter's book `The art of statistics' pages 91-93, or Professor George Barnard's discussion (JRSSB 1971 vol 33 page 377):

"Another way to bring the standard theory to bear is to view the population from which we draw our samples as itself a sample from a super-population. This model of the survey process – which can be a perfectly valid model – perhaps does not receive the attention it deserves in the paper we have under discussion. My limited experience of the use of surveys in the social sciences suggests, indeed, that the super-population model is usually the appropriate one. One is rarely, for example, concerned with the finite de facto population of the U.K. at a given instant of time; one is more concerned with a conceptual population of people like those at present living in the U.K. If one adopts the super-population model, then since a simple random sample of a simple random sample is itself a simple random sample, the problems of inference can be dealt with along classical lines."

1b. Adding to that, it looks like the authors looked at all of the PCR screenings made over a two week period. This means that they looked at the whole population of screenings done during this time period and not a random sample of those screenings. Any differences observed between these groups are different but cannot be considered "statistically different" because there is no real potential for variability inherent in THIS design. These are not a random sample of PCR screenings -- they are all the screenings that happened in the time frame. Saying it's a random sample in time is also false because the authors did not actually randomly select these weeks -- they purposively chose them to see what was going on between the groups.

Please see our comments above about randomisation. We performed an observational study and are not claiming that the data came from a random sample. We chose the core two weeks of study (18/1/20 – 31/1/20) for the following reasons:

i. Prior to these two weeks, only a very small proportion of tests were from the ≥12 days since vaccination group because mass vaccination began on 8^th^ January 2021. Exclusive examination of data earlier to 18/1/20 would have limited our ability to assess the effect of vaccination when an effect may be expected according to the phase III clinical trial.

ii. After these two weeks, a relatively small proportion of tests were from the unvaccinated group, because most people had been vaccinated by then.

iii. The prevalence of infection over the two weeks of study was stable in the unvaccinated group, facilitating simple comparison.

In the text, we have already stated “Here, we analyse data from the two weeks spanning 18th to 31st January 2021, during which: (a) the prevalence of COVID-19 amongst HCWs remained approximately constant; and (b) we screened comparable numbers of vaccinated and unvaccinated HCWs.” Furthermore, as we demonstrate, our analysis is robust to the choice of weeks. As discussed above (‘Randomisation and potential for replicability’), it is possible to regard the data as a random sample from a larger population. One limitation of our study includes the sampling from a single location (Cambridge), which could restrict the generalisability of our results. Nonetheless, we think it is very unlikely that vaccine efficacy will differ substantively between hospitals.

We have now included full week-by-week data for the extended six-week period of study (28/12/2020 – 07/02/2021) as an appendix to the manuscript to enable a full reanalysis of our data by any interested parties, and to highlight why the 2-week period used for primary analysis was chosen. As discussed in response to reviewer #3 point 5, we have also removed this analysis from the main manuscript.

In terms of the reviewer’s comment about examining the whole population of screenings as opposed to a random selection of them, it was necessary to examine the full sample as the proportion of positive tests in the vaccinated group in particular was very small (4/1989 tests). For any reliable estimate of a small proportion, a large sample size is required.

1c. What I would recommend instead of presenting these results as if they were inferential is to put 95% Clopper-Pearson intervals (or some other exact method as the authors choose) and describe these populations rather than comparing them as if they were randomly assigned and present these results as observational and descriptive rather than making them seem experimental and inferential.

In reviewer #3 point (3), the reviewer suggests that there was "no real potential for variability in this design". If this were true, then there would also be no justification for presenting confidence intervals. Both p-values and confidence intervals are statements about what would happen over repeated samples of the data. If confidence intervals are meaningful, then so are p-values. A 95% confidence interval is a random interval that would include the true proportion in (at least) 95% of repeated samples. A p-value is the proportion of the repeated samples where the difference between the proportions in the two (or more) groups is as large as the difference that is seen in the observed data. For our study, repeated sampling could be interpreted in at least three different ways (see ‘Randomisation and potential for replicability’ in our response to reviewer #3 point 1). Whilst we agree with the reviewer’s request to include a discussion of the limitations of the study, we disagree that p-values are inappropriate, and we fail to understand why confidence intervals can be appropriate if p-values are not.

To address the reviewer’s point, we have re-analysed all of the data providing 95% confidence intervals, using Wilson’s method (please see results below). This illustrates that both p-values and intervals calculated using an exact method lead to similar conclusions.

26/3,252 (0·80%, Wilson’s interval 0.55 – 1.17%) tests from unvaccinated HCWs were positive (Ct<36), compared to 13/3,535 (0·37%, Wilson’s interval 0.22 – 0.63%) from HCWs <12 days post-vaccination and 4/1,989 (0·20%, Wilson’s interval 0.08 – 0.51%) tests from HCWs ≥12 days post-vaccination (p=0·023 and p=0·004, respectively; Fisher’s exact test, Figure).

A marked reduction in infections was also seen when analyses were repeated with: (a) inclusion of HCWs testing positive through both the symptomatic and asymptomatic arms of the programme (56/3,370 (1·71%, Wilson’s interval 1.28-2.15%) unvaccinated vs 8/2,018 (0·40%, Wilson’s interval 0.20 – 0.79%) ≥12 days post-vaccination, 4·3-fold reduction, p=0·00001); (b) inclusion of PCR tests which were positive at the limit of detection (Ct>36, 42/3,268 (1·29%, Wilson’s interval 0.96 – 1.74%) vs 15/2,000 (0·75%, Wilson’s interval 0.46-1.23%), 1·7-fold reduction, p=0·075); and (c) extension of the period of analysis to include six weeks from December 28th to February 7th 2021 (113/14,083 (0·80%, Wilson’s interval 0.67 – 0.96%) vs 5/4,872 (0·10%, Wilson’s interval 0.04 – 0.24%), 7·8-fold reduction, p=1x10^-9^).

2. The authors (p2) appear to be presenting analyses where they noticed the largest differences between unvaccinated and 12+ days post-vaccination accompanied by p-values to suggest that these differences possess scientific significance. It is unknown how many different tests the authors carried out and it is a well-known statistical fact that the number of erroneous but significant results is directly related to the number of tests carried out on a dataset. The authors are cherry-picking significant findings and presenting them as if this were a scientific study driven by hypotheses that could actually be tested. It looks, to me, like they used all the data they could and found all the significant differences they could find and present them here. There's no discussion of practical significance -- the fact that less than 1% of health care workers who aren't vaccinated are potentially asymptomatic COVID positive. There's no discussion of the false positive or false negative rate of the PCR test that could be impacting all three groups and there's no justification such a large sample size was explicitly necessary for this investigation. The Fisher's Exact test is easily fooled into producing a significant result simply by inflating the sample size. The authors excessive sample size could be enough to create significance when in fact it may not really be there at all. To me, for example, the fact that almost the same number of people tested positive after being vaccinated as did without the vaccine is a bigger problem than talking about whether they were testing positive relative to when they had the vaccine. Chasing after significance has led to missing this important fact about the vaccinated HCW that are still potentially spreading COVID even though they may think they're safe.

Many thanks for offering the opportunity to clarify our data further. In terms of the points made:

1. Choice of weeks. As described in the response to reviewer #3 point 2, the study period analysed was rationally chosen in order to include comparable numbers of vaccinated and unvaccinated HCWs, over a period in which the prevalence of COVID-19 amongst HCWs remained approximately constant. Furthermore, our analysis was robust to examination of an extended six week period chosen using similar principles. To further address this point and provide full data for analysis by others, full week-by-week data is now presented in the appendix (see also response to reviewer #3 point 3).

2. Practical significance. The practical significance of the results is demonstrated by the magnitude of the risk ratios, which are large. The proportion of asymptomatic screening tests is *expected* to be lower than the proportion of symptomatic tests. Even if fewer than 1% of unvaccinated HCWs tested positive over the course of the period of study, a 75% reduction is clearly extremely valuable in preventing nosocomial transmission to vulnerable patients and other staff i.e. from the point of view of infection control within the hospital, the difference between 0.2% and 0.8% is *highly clinically significant*. We have added a sentence “Nevertheless, the 75% reduction in asymptomatic infection offered by vaccination is likely to be crucial to control nosocomial transmission of SARS-CoV-2.” to the Discussion section to highlight this.

3. False positive and false negative rate. Errors in the test represent a form of measurement error, which, unless the sensitivity and specificity are different in vaccinated and unvaccinated individuals, is likely to cause the estimated relative risk to be slightly biased towards the null hypothesis (i.e. the size of the effect will be underestimated).

4. Sample size. We have discussed the necessity for testing the full, large sample size in our response to reviewer #3 point 2. This is required in order to reliably estimate a small proportion of positive tests. When a large sample is available, we fail to see the justification for discarding part of this sample just to reduce the power of a statistical test to detect small effects. We agree that when the sample size is large, a null hypotheses of no effect can be (correctly) rejected when the effect size is non-zero but small. That is one reason why it is important to present the estimate of the effect size, as we have done here.

5. Reviewer’s suggestion that ‘almost the same number of people tested positive after being vaccinated as did without the vaccine’. The basis for this statement is unclear to us. As stated, 26/3,252 (0·8%) tests from unvaccinated HCWs were positive, compared to 13/3,535 (0·4%) from HCWs <12 days post-vaccination and 4/1,989 (0·2%) tests from HCWs ≥12 days post-vaccination. As discussed above, an effect on symptomatic infections in the original phase III clinical trial was seen from 12 days onwards. The most important comparison is therefore between the proportions of positive tests in the unvaccinated and ≥12 days post vaccination groups. Furthermore, the most important comparison is between *proportions* of positive tests, not total numbers.

6. ‘Vaccinated HCW are still potentially spreading COVID’. We agree that the vaccine is not a panacea and have already suggested that vaccinated HCWs are substantially less likely to become infected spread SARS-CoV-2, as opposed to being completely protected. To emphasise this point, we have added the sentence “This protection was by no means complete, suggesting that continued asymptomatic HCW screening, social distancing, mask-wearing and strict hand hygiene remain vital.” to the end of the discussion.

3. The authors present additional analyses where they extend the time period to include more unvaccinated health care workers in an effort to further inflate the apparent difference between vaccinated and unvaccinated HCW. If there was no demonstration of efficacy of the vaccine then there might be a chance for the null to be true in a situation like this. As it was, we expect the vaccinated group to have less incidence of positive testing, so this contrivance is simply a manufacturing of evidence without any real grounding as to why it's an important "finding." Adding in all the rest of the data since the beginning of the vaccination rollout reinforces my earlier point that this truly is now the whole population and should not be accompanied by a p-value for any reason. These findings are just not statistically significant -- they are what they are. If there's any real practical difference that should be discussed here. In addition, the numbers used in the extended time period do not make sense. They do not indicate which groups are being compared (113/14k what? 5/4.8k what?). This whole thing just doesn't make sense to me and looks like it's just another way to produce a significant-looking p-value without having a real experimental design first.

Our intention in analysing the extended six-week period was merely to determine whether our conclusions were robust across a longer period of time. The weeks included in this analysis met at least one of the two major criteria for selection (i.e. constant prevalence of infection in the unvaccinated group, and/or sufficient tests in the ≥12-weeks post vaccination group to assess the proportion of positive tests).

Nevertheless because this has caused confusion, we have removed the six-week analysis, and have given further details in the form of a week-by-week data over this period in an appendix.

Furthermore, vaccine efficacy against asymptomatic infection was absolutely not pre-supposed. Indeed, to try to answer that key question is exactly why we performed the study! To clarify this point, we have changed the statement “A vital outstanding question is whether this vaccine prevents asymptomatic as well as symptomatic SARS-CoV-2 infection, or converts infections from symptomatic to asymptomatic.”.

Please see our other responses above, which already address other points in this section.

4. I do not agree with the authors' conclusions regarding the value of their "evidence." There is nothing in here to suggest that the vaccination has actually reduced the number of asymptomatic carriers in this population. The rate of positive tests in those vaccinated does not appear to be practically different in the vaccinated and unvaccinated and the only way the authors could make this look like evidence was to manufacture significant p-values through purposive comparisons and sample size inflation. The authors do not rely on sound scientific and statistical methodology to demonstrate their results and present their observational findings as if they were the result of a designed experiment.This is extremely problematic. The pandemic has been raging for more than a year and vaccination is our best chance to controlling the spread and potentiating a return to social normality. There are a tremendous number of problems with vaccination; barriers to vaccination are rooted in great mistrust of the health care system and anti-vaxxers are compelling in their arguments because they can look at manufactured research like this and easily demonstrate its flaws and fallibility. They can easily make the point that if the vaccine really worked then why would they have to make things like this up to show it? They can then suggest that it's evidence of a hidden agenda and we've lost the fight. Fabrication of evidence and manufacturing significant findings to create the illusion of science is not going to help move the vaccine into the population desperately hoping for a return to normal but skeptical about how that will happen when they can't trust the people who should be trying to help them. This research which was supposed to show that the vaccine is working does so in a way that looks like it's really prestidigitation made to fool people into doing something they should want to do but don't.

Please see our responses to similar points above. For the avoidance of doubt, our study design was as follows:

a. Record all screening test results during the relevant period;

b. Measure the proportions of positives in the different groups;

c. Compare the proportions of positives between the groups.

It was therefore an extremely simple, transparent study, which would be inherently difficult to manipulate. We reject in the strongest possible terms the reviewer’s suggestions (both implicit and explicit) that we have fabricated evidence, manufactured significant findings or engaged in any sort of prestidigitation. These are categorically untrue.

Response to second round for reviews:Reviewer #1:In Jones et al., the authors are investigating whether single dose BNT162b2 mRNA COVID-19 vaccine protects from asymptomatic SARS-CoV-2 infection in a HCW cohort. They claim that there is a 4-fold reduction in risk of asymptomatic SARS-CoV-2 infection in vaccinated HCW at >12d post single dose vaccination compared to unvaccinated HCW. The study was carried out in a HCW cohort with documented prior SARS-CoV-2 infection at 7.2% (by positive serology). The B.1.1.7 variant may have accounted for about 90% of infections.This is an extremely important question.

Many thanks for these comments, and for noting that our study addresses an extremely important question.

Major concerns1. Prior infection history. A recent Letter published in the Lancet [Charlotte Manisty, et al. Antibody response to first BNT162b2 dose in previously SARS-CoV-2-infected individuals. Lancet. 2021 Feb 25] showed that HCW with a prior history of mild or asymptomatic SARS-CoV-2 infection made a substantially enhanced response following single dose vaccination with BNT162b2 such that the single dose prime was effectively acting as a boost. In this study, the authors report a 7.2% frequency of prior infection in the HCW cohort being studied. However, the authors do not comment on the frequency of prior natural infection across the three groups studied (unvaccinated, <12d post vaccination & >12d post vaccination). This is important because this could be an important potential confounding factor in the analysis of the study. Prior immunity from natural infection or boosted immunity following single dose vaccination may protect from SARS-CoV-2 infection thus impacting on frequency of PCR positive tests across the three groups. Prior infection history (as documented by serology) needs to be taken into account in the analysis. This data is almost certainly available for the HCW cohort being reported here.

Many thanks for this interesting suggestion. Our manuscript was originally submitted on Feb 21, prior to the publication by Mainsty et al. Nevertheless, we have now compared the frequency of natural infection across the three groups studied. The frequency of prior SARS-CoV-2 infection (as documented by a positive serological test for SARS-CoV-2 prior to 28/12/2020) was similar across all groups (7.1%, unvaccinated; 5.6%, <12 days post-vaccination; 5.7%, ≥12 days post-vaccination) suggesting that this did not confound our observations. We have added this data to our manuscript (lines 75-78).

2. B.1.1.7 variant. The authors should show the frequency of B.1.1.7 variant / WT SARS-CoV-2 across each of the three groups studied (as documented by actual study PCR data during study follow up and not a local average). This data should be available for the HCW cohort being reported here. It is picked up by the PCR test as the ∆69/70 deletion results in failure to detect the S gene target while the ORF1ab and N are not affected. This is how PHE was first alerted to the emergence and spread of the B.1.1.7 variant in the UK. There is no biological evidence presented in this study to show that single dose BNT162b2 protects against the B.1.1.7 variant.

Many thanks for this interesting suggestion. We are certainly aware of the potential for some PCR assays to estimate the proportion of ∆69/70 viruses by S gene drop-out. In fact, only a fraction of the Lighthouse labs and wider PHE regional laboratory network routinely use triple target PCR testing – this reflects the different platforms used between laboratories in the network, an intentional design feature which provides resilience in the UK’s testing capacity. As such, the S gene is not targeted by the platform used in the Cambridge COVID-19 Testing Centre (Lighthouse lab). PCR to identify the B.1.1.7 variant was therefore not routinely available during the study period.

We considered it highly likely that infections amongst HCWs in Cambridge would be representative of viruses circulating in the Cambridge community (and indeed, UK) during the study period. Nevertheless, we have now been able to confirm this by examining Nanopore sequencing data for isolates from asymptomatic HCWs. Over the 2-week period used for the primary analysis, 29/43 viral isolates have been successfully sequenced, of which 24/29 (83%) were from the B.1.1.7 lineage. Across the three groups, 15/19 (79%) sequences from positive unvaccinated HCWs were B.1.1.7, compared to 6/7 (86%) from HCWs <12 days post-vaccination and 3/3 (100%) from HCWs ≥12 days post-vaccination (no significant difference between groups).

We have added data on the overall percentage of sequences that were the B.1.1.7 variant to the manuscript (lines 82-83). We have also added two main authors (Dr. Ben Warne and Professor Ian Goodfellow) who played major roles in the sequencing effort, and additional authors from COG-UK to the group authorship.

3. Lack of follow up beyond 18d. The study can only comment on short term impact of first dose vaccination as the latest time point studied is d18. This is a major deficiency of the study as the second dose of BNT162b2 has been delayed by 12 weeks in the UK and the results described here do not address any later time points. There will have been a proportion of HCW that did not have a COVID-19 vaccine (about 35% of HCW in Leicester for example) and this group could be included in the analysis (for comparison at later time points).

We thank the reviewer for highlighting this point. We previously addressed this same point in response to reviewer #2 point #1 (in their original review). We included a paragraph on study limitations, specifically emphasising that “this study could only examine the short-term impact of single-dose BNT162b2 vaccination”, and re-worded the first sentence of the concluding paragraph: “Our findings provide real-world observational evidence of high level short-term protection against asymptomatic SARS-CoV-2 infection after a single dose of BNT162b2 vaccine…”

Unlike in Leicester, >85% of HCWs participating in our screening programme have now been vaccinated. Combined with the rapid decline in incidence of SARS-CoV-2 infection in the local community, in the last week of data we present (starting 01/02/2021), there were only 3/4282 positive tests overall, offering little scope for analysis of later time points amongst our cohort.

Reviewer #3:Thank you for the opportunity to take a look at the revised manuscript. I have read through the authors' responses to both the others' and my reviews and I have some additional comments:1. Thank you for taking a look at what I had to say about your article and addressing many of the statistical issues I commented on in my initial review. The authors are to be commended for seeking out professional advice and the addition of two professional statisticians to the manuscript says a lot about the authors' commitment to scientific rigor.

We are grateful for this comment, and agree with this point.

2. I would like to thank the authors for the comment regarding p-values and confidence intervals. The authors are, of course, correct that p-values and confidence intervals "come from the same place" whereby changing from one incorrect form of inference to another is not really making a real change. However, the confidence interval provides considerably more information to the reader where the p-value provides no information at all. A 95% confidence interval gives the reader a rough idea of how large or small an effect one may expect to see in repeating a study. A p-value may tell you the chance of seeing a result as extreme in magnitude of occurring in replication but evidence based practice should be more about bang for one's buck. Including the confidence interval, while still inappropriate for inferential purposes in this study, provide greater description of the data and that is an invaluable addition to this paper. Thank you for choosing to make the article more valuable by adding in this very useful information.

We are grateful for this comment, and agree with this point.

3. Following from my previous comment, I see that the authors chose to use Wilson's approach for producing their intervals. My guess is they chose this approach because the Clopper-Pearson approach (an exact approach) gives a wider interval and my guess is that led to "loss of significance." Considering their decision to use other exact tests makes me wonder why they went with an approximation here.

In their prior review, reviewer #3 requested “What I would recommend instead of presenting these results as if they were inferential is to put 95% Clopper-Pearson intervals (or some other exact method as the authors choose)…”

We chose to use Wilson’s exact method at the suggestion of our professional statistical colleagues. Shaun Seaman works at the MRC Biostatistics Unit, and his particular interests are in methodologies for drawing causal inferences from observation data and for handling selection biases. Professor Richard Samworth is the University of Cambridge Professor of Statistical Science and the Director of the University Statistical Laboratory. Both are now included as full authors. They referenced the very highly cited manuscript by Brown, Cai and Dasgupta 2001 “‘Interval estimation for a binomial proportion”. Here, the authors suggest: “The Clopper-Pearson interval is wastefully conservative and is not a good choice for practical use… better exact methods are available…” “For larger *n*, the Wilson, the Jeffreys and the Agresti–Coull intervals are all comparable”.

We have now calculated Clopper-Pearson intervals, and the results are very similar. For example, for our primary outcome 26/3,252 (0·80%) tests from unvaccinated HCWs were positive (Wilson interval 0.55 – 1.17%; Clopper-Pearson interval 0.52 – 1.17%), compared to 4/1,989 (0·20%) tests from HCWs ≥12 days post-vaccination (Wilson interval 0.08 – 0.51%; Clopper-Pearson interval 0.05 – 0.51%).

4. The authors make a point about "estimating a small proportion" requiring a large sample size. This really is not the case at all. Any sample size is sufficient for estimating a proportion of any size. Yes, some samples may give an estimate that will never be exact, but this does not mean that the estimator itself is problematic. Consider a population with 100 individuals and one of them is sick so the true P = 0.01. If we select a sample of size n=5 we will never get a good estimate of the truth from our sample. There are a total of (100C5) possible samples or 75287520. Of those, 71523144 result in an estimate of p=0 and the remaining 3764376 produce an estimate of p=0.2. Clearly, both resulting estimates are wrong but, on average the estimate comes out to be 1% which is the true population proportion. What the authors really mean is that if they want a really PRECISE estimate of their small proportion then they need a really large sample size. An unbiased estimator doesn't get more unbiased as the sample size increases; it just gets more precise, which is what the authors need if they want to show that their really small differences are statistically significant.

We agree with the reviewer that to estimate precisely the small proportions of positive cases in each group, a large sample size was required. This is why we included all screening test results in our analysis, rather than a random sample. We agree that we should have used the term precise rather than reliable in our original response, and thank the reviewer for pointing this out.

5. The authors' response regarding clinical significance is greatly appreciated. I am a statistician and not a clinician, so when I see two percentages that are considerably small it is hard for me to know that they are actually important. I appreciate the clarification as it speaks to the relevance of the data from a practical perspective, something I noted was missing in the original draft. The authors do use percent change to create an inflated perceived improvement which I think is perhaps an opportunity to sensationalize a finding that should already be important if a change from 0.8% to 0.2% is already highly significant from a clinical point of view. I just do not understand why, if these results are so clinically important, why the authors need to paint this vivid picture of statistical significance. It makes me wonder, and a little concerned that this was ok with the statisticians that they added to the paper. The use of ratios and percent change are contrivances of modern epidemiology to create effect when they may simply not be there otherwise. If the practical significance is there be honest about it.Ultimately, the authors' work is very important for many descriptive reasons. My lack of clinical knowledge gives me the inclination to take the authors at their word regarding the practical importance of their findings. However, my considerable statistical knowledge and experience has me wondering.… if these results are so important, why do the authors have to abuse statistical methods to sell them? If they're really this big of a deal, shouldn't they be able to sell themselves on their own merit?

It is certainly not our intention to inflate the perceived improvement, and we agree that absolute differences can be important. We have made precise estimates of the small proportions of positive cases in each group, as acknowledged by the reviewer in point #4, which facilitates assessment of the change in %infection between groups. To address this point, we have changed “Nevertheless a 75% reduction in asymptomatic infection…' to 'Nevertheless a four-fold reduction from 0.8% to 0.2% in asymptomatic infection…”.

By analogy, if the prevalence of COVID-19 infection in two regions of the UK were (for instance) 0.8% vs 0.2%, the reviewer would presumably agree that such a difference would be exceptionally important.

To be completely transparent, we have presented both proportions (as %) and absolute numbers in every instance in the text, together with all the raw data in the appendix. We believe that both provide valuable information and context, and have made great efforts to report the significance of our findings accurately instead of sensationalising them. We therefore do not feel it is at all fair to say that we have 'abused statistical methods'. We hope that on reflection the referee will accept this and retract this unjustified accusation.

If the authors are willing to report their findings descriptively, using 95% Wilson intervals to show effect (as they agree this is important in their response) and eschew p-values then I will re-consider my decision. If the authors present their findings cleanly, without resorting to ratios and percent differences which sensationalize the findings and inflate their perceived importance then I will be happy to accept their work for publication. Finally, I think it's important for all of the authors to understand that statistical methods are about quantifying uncertainty and not about coming up with the best estimate of the truth. The beauty of an observational study is in its opportunity to provide us with an idea of how things are going; from that perspective ANY sample size can provide sufficient information to inform future research. It's only when we're blinded by the need for statistical significance that we start looking for ways to "find results" rather than answer real questions.

As in our original rebuttal, we believe that p-values also have merit alongside confidence intervals, whilst accepting the reviewer's point that additional information is provided by confidence intervals. We have therefore now included both, using identical wording to the statement we provided in our original rebuttal. We note that the reviewer accepted this statement as “an invaluable addition to this paper” (reviewer #3 point 2 above; lines 53, 54, 55, 62, 63, 65).

For the reasons stated above, we believe it is clearly helpful to include ratios (fold reductions) to estimate the size of the effect. This is entirely consistent with the approach taken by other studies in the medical literature. Again, we absolutely reject the reviewer’s suggestion that we have attempted to “find results”. This is simply untrue.

If the editors wish us to remove these ratios, we would be willing to do so, however we believe this would make life unnecessarily difficult for the reader.

I do hope that my comments are helpful and give the authors an opportunity to decide how to best convey the story their data are telling.